# Correlation between Friction and Wear in Cylindrical Anchorages Simulated with Wear Machine and Analyzed with Scanning Probe and Electron Microscope

**DOI:** 10.3390/ma16051991

**Published:** 2023-02-28

**Authors:** Tomasz Dąbrowa, Dominik Badura, Bartosz Pruchnik, Ewelina Gacka, Władysław Kopczyński, Marcin Mikulewicz, Teodor Gotszalk, Edward Kijak

**Affiliations:** 1Department of Prosthodontics, Wrocław Medical University, ul. Krakowska 26, 50-425 Wrocław, Poland; 2Nanometrology Department, Wroclaw University of Science and Technology, Janiszewskiego 11/17, 50-372 Wroclaw, Poland; 3Department of Dentofacial Orthopaedics and Orthodontics, Division of Facial Abnormalities, Wrocław Medical University, ul. Krakowska 26, 50-425 Wrocław, Poland

**Keywords:** AFM, biomaterials, dental wear

## Abstract

This paper presents the possibilities of applying atomic force microscopy (AFM) techniques to the study of the wear of prosthetic biomaterials. In the conducted research, a zirconium oxide sphere was used as a test piece for mashing, which was moved over the surface of selected biomaterials: polyether ether ketone (PEEK) and dental gold alloy (Degulor M). The process was carried out with constant load force in an artificial saliva environment (Mucinox). An atomic force microscope with an active piezoresistive lever was used to measure wear at the nanoscale. The advantage of the proposed technology is the high resolution of observation (less than 0.5 nm) in the three-dimensional (3D) measurements in a working area of 50 × 50 × 10 µm. The results of nano wear measurements in two measurement setups are presented: zirconia sphere (Degulor M and zirconia sphere) and PEEK were examined. The wear analysis was carried out using appropriate software. Achieved results present a tendency coincident with the macroscopic parameters of materials.

## 1. Introduction

Prosthodontic treatment with overdentures increases the comfort of removable dentures while restoring extensive missing teeth. They may be anchored on the patient’s teeth or intramedullary implants. Satisfactory retention can be achieved with telescopic crowns, among others. These are prosthetic restorations that transfer masticatory forces parallel to the long axis of the teeth to the periodontium and alveolar bone, resulting in better stabilization in the alveoli and better durability of the prosthesis [1]. The simplest telescopic abutment is an assembly of two components: an inner crown called a patrix and an outer crown called a matrix [2]. The retention of cylindrical telescopic crowns is based on the frictional force between the patrix and the matrix. During use, the abutment surfaces come into proximity and make contact at the molecular level. As the material wears away during chewing, the anchorage loosens and the unit pressure value decreases. The presence of saliva usually results in mixed friction in telescopic joints. The total friction force Ft is the sum of the dry friction FD, the boundary friction FB, and the fluid friction FL. Based on our research, we can conclude that the retention force of a telescopic prosthesis can be caused not only by friction but also by the force of adhesion, the type of biomaterials used to make the telescopes, and the properties of human saliva [3,4].

Based on the literature data, we believe that in dental cylinder crowns, many types of material combinations are applied, from which primary and secondary crowns are produced. Therefore, there is no one preferred application. Chosen material pairs were examined, and they were successfully applied in the manufacturing of telescopic crowns in the prosthodontic department. Gold alloys and PEEK are used for primary crowns, while zirconia is for secondary ones [5,6]. Based on the literature, it can be found that either gold alloy or PEEK (which are relatively soft materials) can be combined with material as hard as zirconia [4]. Therefore, primary crowns have to be polished in a water jacket in order to provide contact with the sufficient contact surface between the patrix and matrices of telescopic contact, securing the retention stability. We believe that retention may also be influenced by a series of interactions, such as adhesion between surfaces, fitting of elements, materials, and medium (in the dental case, saliva).

In classical testing methods, good coordination of the components of the mechanical assembly is required. In our opinion, the use of a probe with a spherical end provides contact with a constant average cross-sectional area with the tested surface, provides better freedom in assembling the elements of the system, and does not require pre-lapping. The undoubted advantage of the presented method is the lack of calibration of the system. The use of a spherical probe results in the formation of modifications in a relatively small working field, which makes it possible to carry out research at the microscale, where high-precision methods such as AFM and XRD can be used.

### Aim of Study

The aim of this study was to illustrate methods and techniques for describing friction and wear phenomena using specialized measurement systems that allow microscopic examination of material transfer between mating components of selected biomaterials used in telescopic anchorages. 

## 2. Materials and Methods

### 2.1. Contact Mechanics

Between nonideal (incohesive and uneven) surfaces, a great variety of phenomena occur. In general, most of the processes tend to be destructive interactions, as they lead to impairing the properties of at least one of the working surfaces, the properties being: hardness, roughness, and flatness (or another type of shape tolerance). In critical cases, rapid wear can lead to the destruction of a specific surface, especially in cases of surface engineering (e.g., coating). The quantitative assessment of surface condition in correlation to wear process parameters is an important tool in contact mechanic examination.

For the effect of wear, two main processes can be named. Ductile deformation is not inherently damaging, as it does not cause the removal of material, and as such, it will not lead to the degradation of surface layers, which is especially important for coatings. Yet it impedes the properties of the object, altering the geometry and roughness. Ductile materials are especially susceptible to this kind of degradation. The friction of surfaces almost always leads to abrasion. This process is destructive, as due to friction, materials lose cohesion. Loose particles are a factor that accelerates the abrasive process even more. The removal of loose fractions leads to the reduction of mass. Abrasion tends to lower the roughness of surfaces. Both processes can coexist, and their proportion relies mainly on material composition. 

In experiments, lubricated contact was examined. The introduction of boundary lubrication reduces the friction coefficient, thereby reducing abrasion. In the border case of sliding contact, abrasive wear is no longer present, as lubrication is an intermediate layer. The introduction of lubricant, however, leads to a few more surface effects. Apart from the obvious effects of chemical reactivity, viscous slip contact enhances erosive effects, the main one of which is cavitation. Some models [7] predict the influence of cavitation on the erosion of the contact surface. Less cohesive materials are also susceptible to decohesion due to pressure alterations. In such a case, two well-established mechanisms must add a third.

In contrast to plastic deformations, erosive phenomena would also cause the removal of material (or at least its decoherence in the subsurface region). In contrast to abrasion, it could also increase the roughness of the surface. 

Wear between bodies is highly dependent on material properties, the geometry, and the roughness of surfaces. Contact is always constituted by two bodies; therefore, in an examination, contact pairs must be examined. Information about the process can be derived from a few basic facts about the surface, mainly its topography and roughness. It is necessary to use appropriate tools that can deliver precise values for those quantities.

### 2.2. Wear Machine

In our experiments, we model the interactions between the primary and secondary crowns in a telescopic denture with a sphere probe and a test specimen setup, in which the load force is well-defined and known. In the developed system, the probe moves in a reciprocal manner to simulate interactions that occur during the removal and insertion of the prosthetic restoration. This corresponds with our previous model experiments, in which friction between the abutments of telescopic dentures was measured [4]. The following compositions of prosthetic materials were used for testing: gold alloy Degulor M (zirconium dioxide and PEEK (polyetheretherketone)) and zirconium dioxide. Degulor M is a precious dental alloy with a high gold content. It is used for milled work in the telescopic crown technique. Zirconium dioxide is also used in dental prosthetics in the water-jacketed milling process. It is a very hard material. Its hardness is 13 GPA on the Vickers scale. PEEK is a polymeric material that has been used in medicine for many years. In dentistry, it is used to make dentures. It is a biocompatible material with a modulus of elasticity similar to human bone. In the first stage, specimens were prepared according to the procedures used when making prosthetic restorations for patients. The samples were made using the lost-wax technique. The wax pattern was milled with a dedicated cutter. After the wax was converted to metal, the surface of the specimen was machined to be coarse and fine. For both processes, we used Bredent cutters with a diameter of 2.35 mm. To smooth the surface, we used Bredent’s milling oil (REF 550 0000 8). The process parameters were chosen according to the recommendations of the cutter manufacturer and our clinical experience. Our experiments were performed in a Mucinox artificial saliva solution (Parnell Pharmaceuticals Limited, Dublin, Ireland). The Mucinox solution contains water, sorbitol, xylitol, Eriodictyon Crassifolium (Yerba Santa), natural lemon flavoring, ascorbic acid, sodium benzoate, and citric acid. This remedy also contains natural mucin extracted from the Yerba Santa plant. It forms a natural protective layer on the oral mucosa. In addition, the preparation binds water and transports it to the cells of the mucosa, thus further alleviating discomfort. It therefore mimics all the functions of natural saliva. Thanks to the presence of citric acid and ascorbic acid, which regulate the acidity of the preparation, and sodium benzoate, which prevents the growth of microorganisms, it is possible to carry out tests under constant conditions and compare measurements on different surfaces. In order to be able to determine the wear of the components of the telescopic abutment systems, the prepared samples were subjected to friction in a device designed for this purpose (Figure 1), in which the pressure was modeled with reference weights [8,9,10,11].

The test samples were placed in the bracing forceps in a dish filled with a Mucinox artificial saliva preparation (Parnell Pharmaceuticals Limited, Ireland). The vessel, after being attached to the movable table of the device, was set into a reciprocating motion with a preset travel range (up to 5 mm) and a table travel speed of 0.5 mm/s. The device was controlled with a PC application developed in the LabVIEW environment (Figure 2) and connected to the control board via the USB interface. To achieve high precision in movement, a micrometer head and a stepper motor (with 200 steps/revolution) with an integrated planetary gearbox (with a 5.18:1 ratio) were used in this device. The number of steps of the motor, corresponding to a given displacement of the sample, was determined using the following equation:(1)$$n=\frac{Lx\;\cdot i\;\cdot SMres\cdot DRVres}{µMres}$$
where *n* is the number of steps of the stepper motor corresponding to the preset displacement, *Lx* is the preset displacement (mm), *i* is the planetary gearbox ratio, *SMres* is the number of steps per revolution of the stepper motor in full-step mode, *DRVres* is the step resolution of the stepper motor driver, and µ*Mres* is the resolution of the micrometer screw in millimeters.

A zirconium (Zr) ball with a diameter of 30 mm was used as a wear probe, attached to a vertical arm suspended over the test specimen. The zirconium ball was manufactured by MERAZET [12]. The arm was loaded with a mass not exceeding 150 g. Each of the prepared materials was subjected to a wear process lasting no less than 5000 operating cycles (where 1 cycle = 1 table pass in a given direction) [8,13,14].

Before and after the actual friction and wear measurements, the fabricated specimens were analyzed using an atomic force microscope (AFM) and a scanning electron microscope (SEM). Material transfer between the mating components may be identified by X-ray spectroscopy (EDX), which identified the elements on the tested (mating) surfaces.

### 2.3. Atomic Force Microscopy

In our experiments, we used an atomic force probe microscope operating with the so-called active piezoresistive cantilevers to study the wear in dental biomaterials. The active piezoresistive cantilevers integrate with the spring beam, a piezoresistive deflection detector, and an electrothermal deflection actuator (Figure 3) [15]. In this way, the cantilever deflection and the excitation of the beam resonance vibration can be controlled electronically, which reduces susceptibility to various drifts and errors introduced by laser optics. The sensitivity of the piezoresistive deflection detector is in the range of tens of microvolts per nm [16,17], which enables the application of the active piezocantilevers [18] in contact and non-contact surface imaging. The entire microscope head is presented in Figure 4. It operates in the so-called scanning probe mode, which means that the cantilever tip moves over the immobile sample. Such an architecture makes it possible to investigate technological specimens of large dimensions. The piezoelectric scanner enables cantilever displacement in the X and Y directions in the range of up to 100 microns. In the Z direction, the cantilever can be moved up to 10 microns. The measurement head is placed over the investigated sample on two micrometer screws. The third one is controlled by a DC motor, which serves to adjust the distance between the surface and the tip. The scanning field is observed using an endoscopic CCD camera. The measurement head, which integrates a piezoresistive preamplifier, is connected to a scanning controller. In the performed experiments, the surface was scanned at a frequency of 2 lines/s, and 256 lines were recorded in the X and Y directions.

The described system enables surface scanning with a contact force ranging from tens to hundreds of nanonewtons. In this way, not only precise surface imaging but also surface modifications are possible. In our opinion, the experiments performed on the micro- and nanoscale can be scaled to the macroscale. The undoubted advantage of such measurement is that a relatively small sample of material is enough to measure and evaluate the roughness of the dental material before and after the wear experiment.

Based on the AFM measurements, it was possible to localize precisely the area affected by the wear process. The first series of surface measurements were performed before the wear test process of the sample in order to eliminate the influence of other factors, such as chemical surface modification. Subsequently, the sample was scanned in the wear area to determine the surface roughness outside and inside of the modified area and observe the influence of the sphere probe on the tested specimen (Figure 5). The recorded images were analyzed using Gwydion software.

### 2.4. Scanning Electron Microscopy

A scanning electron microscope (SEM) with a focused ion beam (FIB) was used to investigate the topography of the sample before and after ZrO2 sphere wear tests. However, an energy dispersive spectrometer (EDS), which is integrated into the microscope, was used to analyze the chemical composition of the sample after wear tests to identify the residual ZrO2 sphere material on the sample surface.

The SEM FIB microscope (FEI Helios NanoLab 600i) is a multifunctional device. The electron beam is applied for imaging (using detectors connected to the microscope) and deposition of material (using a gas injection system (GIS)). The gallium ion beam is similarly used for imaging and deposition but also for milling and ion implantation into the sample material. The imaging and prototyping of new objects are performed on a single nanometer scale. The possibility to characterize the chemical composition of the sample is provided by an EDS (from EDAX, Mahwah, NJ, USA), connected to the vacuum chamber of the SEM FIB microscope. The interaction of the scanning electron beam with the sample elements causes a knockout of electrons from their atomic shells. Electron vacancies are created and are willingly occupied by electrons from higher shells, which transfer their excess energy in the form of X-rays. The energy of the X-rays is related to the energy difference between the shells of the atoms, allowing them to be identified. Then, the X-rays pass through a window in the EDS into a silicon drift detector (SDD), cooled by a Peltier module to −30 °C to reduce the noise ratio. X-rays generate electron-hole pairs in a Si crystal (energy required to create one electron-hole pair: 3.8 eV; X-ray energy of, for example, Zn Kα: 8.63 keV). Charges are separated from each other by biasing the crystal. The characteristic X-ray line is identified and assigned to the element by detecting the height of the electrical signal by the Team software [8].

## 3. Results and Discussion

Successful tests were performed to simulate the conditions in the oral cavity during clenching, using a wear machine developed by the article’s authors. In this way, a series of measurements were conducted under controlled pressure (load force) in the artificial saliva environment. In the performed investigations, we studied the wear of the Degulor M gold alloy and PEEK specimens.

The developed measurement method makes it possible to study the interactions at nanometer resolution. The recorded results can be scaled up to ten microns, which in our assessment corresponds with the real conditions in the oral cavity. Based on the performed investigations, we found that the roughness coefficient Ra, defined as the arithmetic mean of the deviation from the mean value of the surface height of the Degulor M material, decreased from an average value of about 11 nm to a value of about 4 nm after the wear (Figure 6).

In addition, after the precise localization of the area where the model interactions between the sphere and the test specimen were carried out, characteristically arranged lines resulting from the friction of the zirconia penetrator on the surface of the material were observed. It was also seen that a characteristic material accumulation had formed on the edges of the worn area, which indicated a similar behavior of the structure as during the rolling process. In this case, the material, in order to maintain the same volume, caused boundary deformations of much greater height relative to the simulated area or the one outside it.

A PEEK specimen was another sample analyzed in our experiments, which were performed under the same conditions as the measurements on the Degulor M specimen. The area before wear exhibited a roughness Ra of about 72 nm (Figure 7). After the wear procedure, the calculated roughness was 180 nm (Figure 8). Based on the surface analysis before the wear process, it was noted that the zirconium penetrator encountered more friction than the Degulor M sample. The zirconium oxide microsphere, as the hard probe, tore up some fragments of the flat specimen, creating micro-pits and irregularities. The introduced micro-strains were also observed, as the obliterated area was clearly deformed and the micro-cavities were clearly stretched along the obliteration line.

In this way, the wear resistance of the prosthetic biomaterials was successfully characterized. The developed methods, combining the wear machine and the AFM technology, required very small material samples, and the results obtained at the nanoscale were scalable to macroscopic dimensions.

Due to the ductility of PEEK, the worn rift has greater dimensions than the AFM scan area. The green arrow indicates the same rift on the PEEK specimen.

## 4. Conclusions

In the performed experiments, the wear properties of the Degulor M and PEEK specimens interacting with the zirconium oxide sphere were analyzed. In order to analyze the wear phenomena, we applied AFM advanced technology based on the active piezoresistive cantilevers. This technology makes it possible to investigate technological specimens, which are formed quite often by the details of a larger height. It should be noted that in our experiments we analyzed and compared the same surface areas prior to and after conducting the wear process, which makes the presented results reliable and accurate. The recorded wear properties were different for the ductile polycrystalline alloy Degulor M and the ductile but amorphous PEEK polymer (Table 1).

Modifications of the first pair were dominated by ductile deformation. The worn area was about 21 μm in width. The ductile deformation is clearly observable in a form of the aggregated material on the sides of the wear trail. Reduction in volume of the surface is omittable, therefore it can be stated that Degulor M was deformed and not abrased. A significant reduction of roughness is also present in the worn area. The material was locally smoothened, possibly reducing its susceptibility to further wear-driven modifications.

Analysis of the second material pair exhibits also most certainly undergone ductile deformation. The exact width of the worn area was not established, as its dimensions were much greater than the working area of the AFM system. However, there is no distinguishable aggregation of material removed by the indenter, leading to the conclusion that the material is abrasively removed from the surface. The mass of the substrate is reduced, which means, in the case of the actual joint, that the parameters of the fit can change. Moreover, there is no observable reduction in roughness, which leads to the conclusion that the presence of the erosive processes coincides with the abrasive removal of material. As the abrasion intensity is enhanced by the surface roughness, an increase in the wear induced by a kind of positive feedback must be observed.

We showed that between two examined pairs of materials, the set Degulor M—zirconium oxide is much more well-fitted. Material removal is hardly measurable, while the reduction of roughness improves the wear resistance.

It has also been shown that AFM measurements are well-suited for examination of the local wear phenomena. Reduction of the affected area is scalable to the macro-dimensions while enabling measurements of the surface parameters with a resolution down to fractions of a nanometer. Further improvement of the method involves a further reduction of the wear process dimensions, possibly to the scale at which multiple experiments can be performed within the range of a single AFM measurement. This step is especially hard to perform in the case of yielding materials, which require especially small indentation forces for accurate measurements. Using an AFM cantilever operating as a nanomodification and measurement tool can be the solution.

## Figures and Tables

**Figure 1 materials-16-01991-f001:**
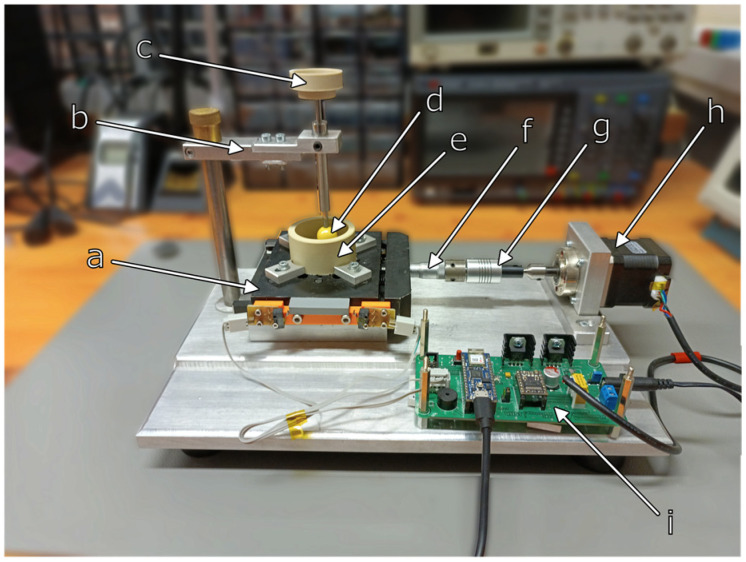
Wear machine: a—sliding table, b—indenter crane, c—indentation mass holder, d—indenter effector (zirconium ball), e—sample fluid cell, f—micrometer screw, g—clutch, h—stepper motor, and i—stepper motor driver.

**Figure 2 materials-16-01991-f002:**
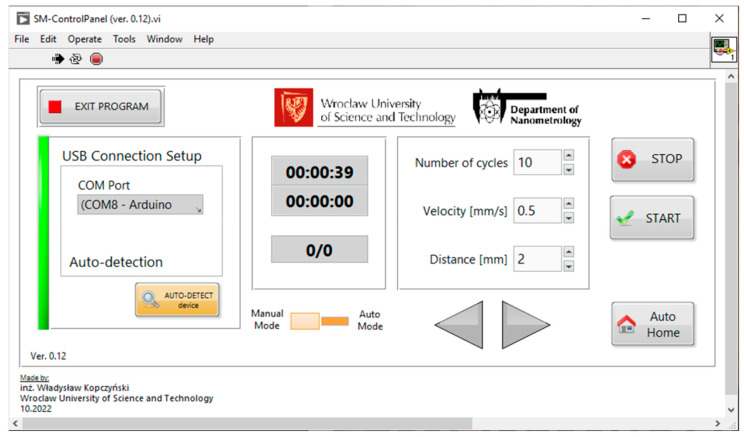
Software control panel for the definition of parameters of the wear study.

**Figure 3 materials-16-01991-f003:**
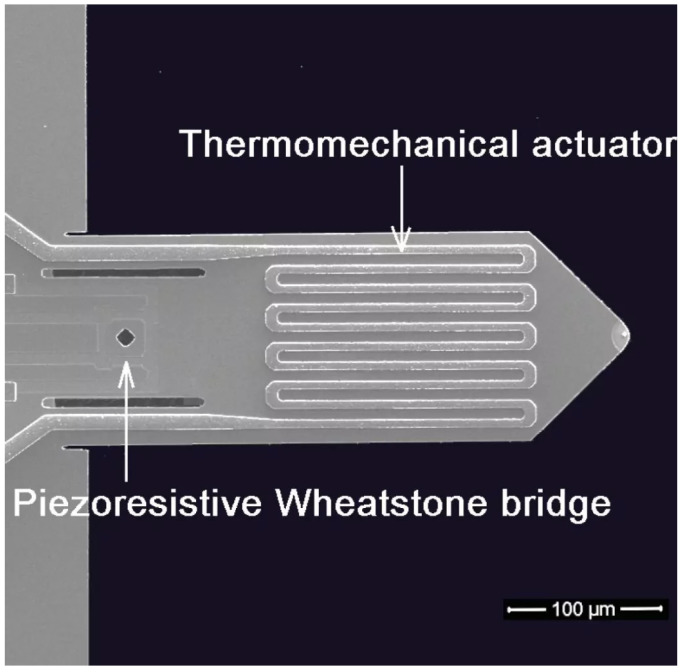
Active piezoresistive cantilever used in the conducted experiments. Cantilevers purchased from NanoAnalytic GmbH. Picture taken from the product website.

**Figure 4 materials-16-01991-f004:**
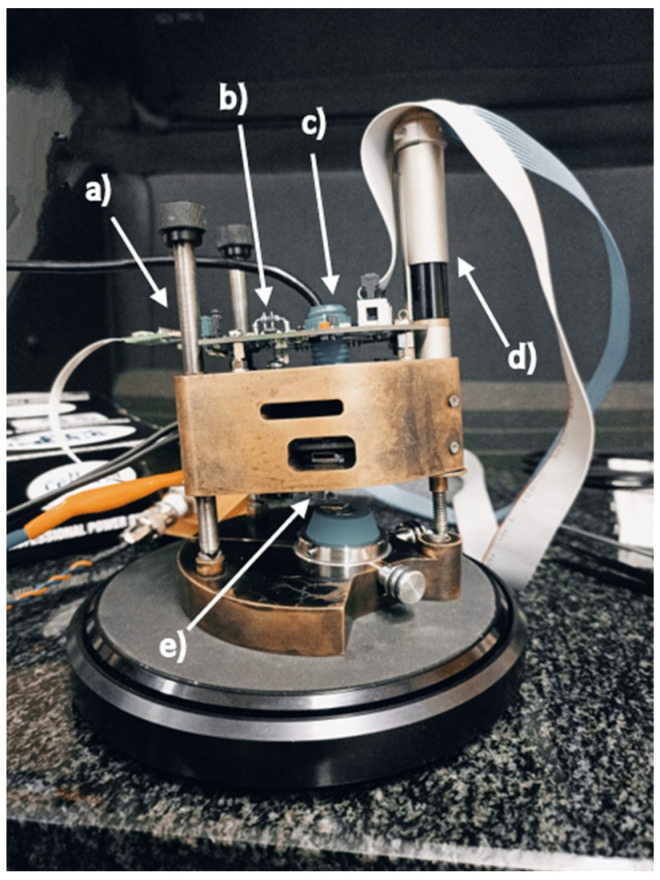
Atomic force microscope head operating with an active piezoresistive cantilever used to study the wear properties of the model prosthetic dentures. (a) Micrometer screw adjusting the height over the investigated sample; (b) signal preamplifier; (c) endoscopic camera holder; (d) DC motor for tip-surface coarse approach; and (e) Cantilever holder.

**Figure 5 materials-16-01991-f005:**
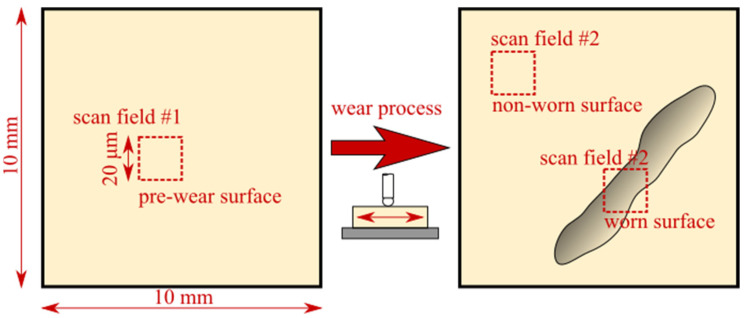
AFM scan fields throughout the wear process over the area of the sample (not in scale).

**Figure 6 materials-16-01991-f006:**
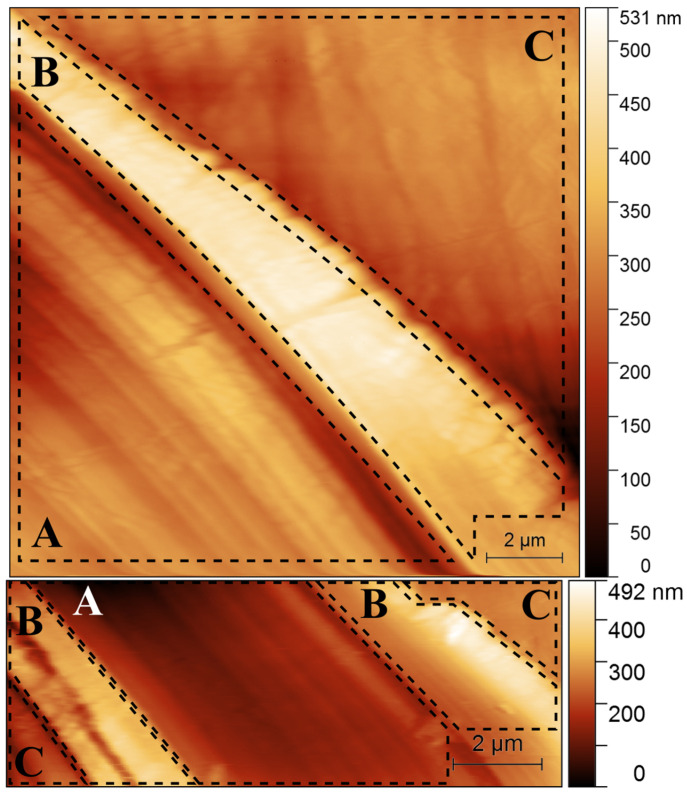
AFM pictures of Degulor M surface after modification. A worn rift of about 15 um in width (area A), aggregation of worn material (area B), and unmodified material (area C). In the cross-section, an approximate shape of the zirconium oxide sphere probe intending the specimen is presented.

**Figure 7 materials-16-01991-f007:**
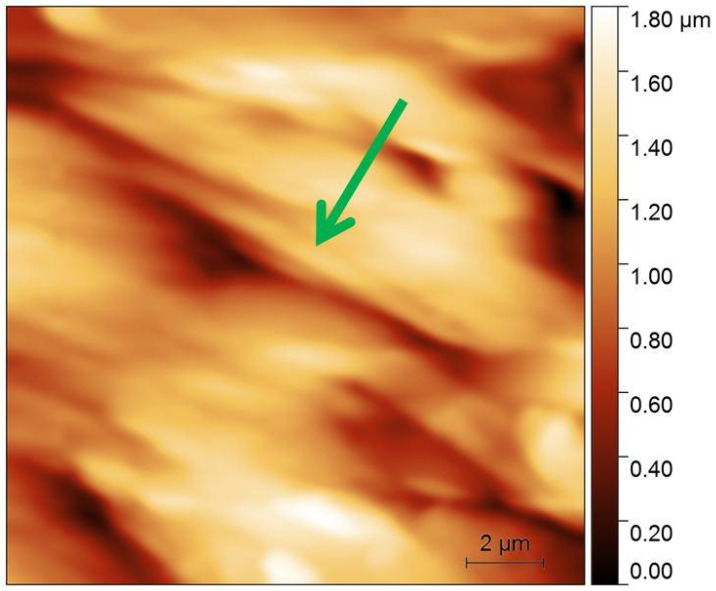
AFM pictures of PEEK surface before modification. Arrow points onto the same surface detail as in the next picture.

**Figure 8 materials-16-01991-f008:**
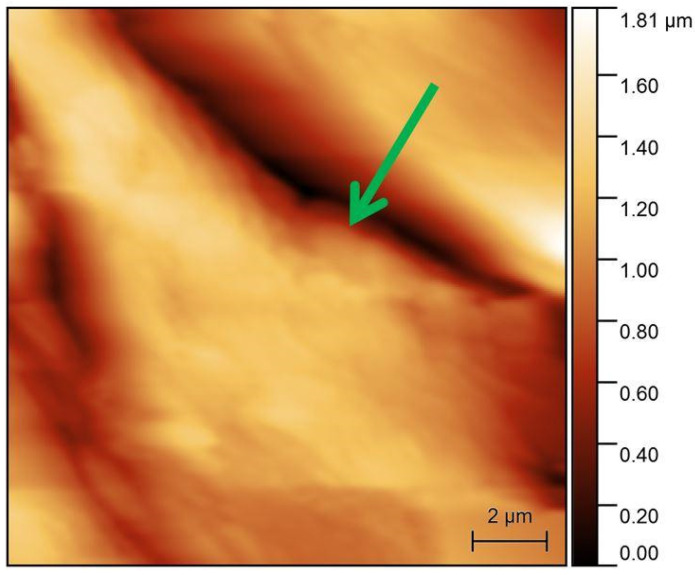
AFM pictures of PEEK surface after modification. Arrow points onto the same surface detail as in the previous picture.

**Table 1 materials-16-01991-t001:** Comparison of roughness before and after modification.

Material	Roughness before Modification [nm]	Roughness after Modification [nm]
Degulor M	11	4
PEEK	72	180

## Data Availability

Experimental data from AFM measurement are available for analysis on webpage under address: https://drive.google.com/drive/folders/1EDz-ZZcCwk2qEi9tnA15kdJ903_skkYZ?usp=sharing (accessed on 22 February 2023).

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
