# Peer review of "Correlation between Friction and Wear in Cylindrical Anchorages Simulated with Wear Machine and Analyzed with Scanning Probe and Electron Microscope"

_materials, 2023, doi:10.3390/ma16051991_

Round 1

Reviewer 1 Report

Dear Authors,

your work is very interesting, however I suggest to improve the readability of your manuscript. in particular, you should improve the introduction by including more information about the systematics studied and the rationale for conducting this work. 

To improve the understanding of the results I would suggest that you summarize them in a dedicated paragraph ("Results") and then conduct their discussion/explanation within the "Discussion" section

Author Response

Dear Reviewer,

I am enclosing my responses.

Yours sincerely

Reviewer 2 Report

The manuscript entitled “Correlation between friction and wear in cylindrical anchorages simulated with wear machine and analyzed with scanning probe and electron microscope” investigated the effectiveness of a new wear test method using AFM technique. The authors concluded that achieved results presented a tendency coincident with macroscopic parameters of materials; however, what were achieved results and macroscopic parameters are not clear. It is necessary to brush up this manuscript before publication. At least following points should be revised.

The reviewer feels misleading the title of the manuscript because the authors did not directly evaluate friction in this study.

The authors described the retention of cylindrical telescopic crowns, but the experimental set-up was apart from telescopic crowns.

Moreover, the combinations between zirconia and gold alloy or that between zirconia and PEEK are not popular for telescopic crowns. The authors should explain why these combinations were selected in this study.

If the authors wanted to suggest usefulness of the author’s suggested method, the authors should evaluate merits and demerits of the new method comparing with previous methods.

If the author considered to submit as original article, the authors should describe research hypothesis and containing a control group. In the present style, it might be possible to be published as a short communication.

Technical terms of the telescopic denture crown are not same in line 35 (partrix and matrix) and line 86 (primary and secondary).

Line 95   A semi-sintered zirconia blank is usually used with dry milling. This description of water-jacketed milling should be revised.

Line 99  Regarding a modulus of elasticity, it is necessary to specify the type of human bone. It might be the modulus of elasticity of cancellous bone similar to that of PEEK.

Figure 1  Please change the color of letters in Fig.1 because blue font is difficult to understand.

Line 134  What is µMres?

Line 142  It is a zirconia (ZrO2) ball. Please describe the product name and manufacturer.

Line 143  The load should be more specific.

Line 177  It is necessary to describe how to determine the surface coeddicient (Ra) in this study

P193  The manufacturer of “Gwydion software” should be described.

P207  "Clenching" is not with moving action. The load is too small compare the occlusal force; therefore this test method did not simulate the oral cavity during clenching or mastication

Line 216   12 nm and 7 nm. These values are not same as in Table 1.

Line 220   It is not zirconium but zirconia.

Line 228   80 nm and 160 nm  These values are not same in Table 1.

Reference  According the instruction for authors, journal articles should be describes as follows;

Author 1, A.B.; Author 2, C.D. Title of the article. Abbreviated Journal Name Year, Volume, page range.

For example of Ref 1 is;

Wöstmann, B; Balkenhol, M; Weber, A; Ferger, P; Rehmann P. Long-term analysis of telescopic crown retained removable partial dentures: survival and need for maintenance. J Dent 2007, 35, 939-45.   (some authors family names in the manuscript are incorrect.)

Author Response

Dear Reviewer,

Thank you very much for your comments, they were very substantive. I am enclosing my responses.

Yours sincerely

Reviewer 3 Report

Scanning electron microscopy 196

Authors have included AFM, one of the advanced techniques. Results need to be shown and discussed regarding SEM/FIB/EDX. Please add SEM/FIB/EDX to complete the information.

Author Response

(The authors gave the same response as above.)

Reviewer 4 Report

The work is appropriate, using a novel technology, which the authors describe properly, with very specific objectives and results in line with these objectives. But it seems that it has only been a test, which should make it clear that it is a pilot test, and therefore they are not consolidated results.

Although the journal is a materials journal and the study is very focused on the technology used to meet the stated objectives, the authors should also discuss the possible clinical application of their results, because although it is intuitive, they have not taken into consideration all the conditioning factors, in particular the important difference of using metal or PEEK in the oral environment in terms of the possible corrosion or release of ions from the metal which is being subjected to mechanical stress.

Author Response

(The authors gave the same response as above.)

Round 2

Reviewer 1 Report

Dear Authors,

Great job, now I think that your work is suitable for publication 

Reviewer 3 Report

Thanks, I found the corrections sufficient for this work.